# Fabrication of In Situ rGO Reinforced Ni–Al Intermetallic Composite Coatings by Low Pressure Cold Spraying with Desired High Temperature Wear Characteristics

**DOI:** 10.3390/ma16093537

**Published:** 2023-05-05

**Authors:** Zhikai Liu, Weiqi Lian, Cansen Liu, Xiaohua Jie

**Affiliations:** 1School of Material and Energy, Guangdong University of Technology, Guangzhou 510006, China; 2College of Mechanical and Electrical Engineering, Guangdong University of Petrochemical Technology, Maoming 525000, China

**Keywords:** cold spraying, graphene oxide, heat treatment, wear resistance

## Abstract

In this study, the surface of aluminum powder was uniformly coated with in situ reduced graphene oxide (r-GO) sheets (Al/r-GO). The Ni powder, Al_2_O_3_ powder, and Al/r-GO powders were mixed uniformly in a mass ratio of 20:6:4. In situ rGO-reinforced Ni–Al intermetallic composite coatings were successfully prepared using low-pressure cold spraying and subsequent heat treatment. The microstructure and phase of the composite coatings were characterized using X-ray diffraction (XRD), Raman spectroscopy, and scanning electron microscopy (SEM). The high-temperature wear test was conducted at 200 °C, 400 °C, and 600 °C to understand the mechanism. The results indicate that the in situ rGO-reinforced Ni–Al intermetallic composite coatings exhibit a 33.3% lower friction coefficient and 26% lower wear rate in comparison to pure Ni–Al intermetallic coatings, which could be attributed to the generation of an easy-shearing transferred film between the coating and grinding ball.

## 1. Introduction

Recent studies have demonstrated that aluminum-rich alloy coatings exhibit excellent high-temperature oxidation resistance and anti-wear properties [1,2], making them widely applied in the aerospace and automobile industries [3]. Traditional high-temperature oxidation-resistant coatings mainly rely on ceramic coatings such as SiO_2_ and Al_2_O_3_. However, their high hardness makes them prone to peeling and cracking in cold states, despite performing well at high temperatures. The development of nickel–aluminum intermetallic compound coatings has resolved this issue by maintaining good performance at high temperatures while also possessing good toughness and ductility, making them better suited for complex working environments [3,4]. Amin Amiri Delouei et al. [5,6] investigated the influence of the most significant parameters in the heat conduction of multilayer and composite materials, such as boundary conditions, internal heat sources, and thermal contact resistance, on the complexity of the results. The Ni–Al intermetallic coating is a promising high-temperature protective coating that exhibits good resistance to high-temperature oxidation and corrosion [7,8]. Li et al. [9] produced Ni–Al intermetallic coatings with a thickness of 35 μm on steel substrates through the electro-deposition of nickel and aluminization. However, electro-deposition of nickel is a highly demanding process, making it challenging to control the coating’s thickness and uniformity to meet the requirements of producing thicker coatings for further applications. Xiang et al. [10] investigated the long-term oxidation process of Ni_2_Al_3_ coating and the factors that influence the coating’s service life under high-temperature conditions at 650 °C.

Based on previous studies, the preparation of Ni–Al intermetallic coatings is mainly conducted through several steps, including electroplating of Ni and subsequent aluminization, as well as heat treatment. Cold spraying is a spraying technique based on aerodynamic principles, in which high-speed airflow drives the powder to impact the substrate at critical or supercritical velocity. The sprayed powder particles undergo plastic deformation along with deposition of coatings on the substrate, resulting in a high bonding force and low porosity [11]. This technique offers the advantages of fast coating deposition. Wang et al. [12] demonstrated the feasibility of preparing Al/G protective coatings on magnesium alloy via the low-pressure cold spraying of reduced graphene oxide-coated aluminum powder, which exhibits good corrosion resistance and a strong repassivation ability. Additionally, K. Spencer et al. [13] successfully fabricated NiAl_3_–Ni_2_Al_3_ intermetallic composite coatings on the AZ91 alloy matrix via cold spray coatings along with heat treatment while investigating the relationship between heat treatment time and the porosity of the coating. Thus, facile fabrication of Ni–Al intermetallic coatings via cold spray is highly feasible.

Graphene is a two-dimensional material with unique properties such as high strength, low surface energy, and low shear strength. In recent years, there has been an increasing amount of research on the self-lubricating material graphene, and it has been applied in the preparation of new metal-based self-lubricating composites [14,15]. Due to its extremely high specific surface area, graphene has a significant effect on enhancing the toughness and reinforcement of coatings [16,17,18]. Numerous studies have shown that graphene, as a lubricant additive, can significantly improve the anti-wear and anti-friction properties of materials [19]. To date, scholars from both domestic and international institutions have successfully prepared graphene-reinforced metal-based self-lubricating materials through powder metallurgy, electroplating, and other methods, obtaining excellent wear and friction reduction properties [20,21]. However, research on using a cold spray coating to prepare graphene-reinforced metal-based self-lubricating composites is still limited. Additionally, studies on the high-temperature wear properties of Ni–Al intermetallic coatings are still relatively rare. Therefore, it is of great scientific significance and engineering value to develop practical methods for Ni–Al intermetallic coating preparation and further explore the mechanism of high-temperature wear properties for Ni–Al intermetallic coatings to extend their engineering applications in high-temperature fields.

In this study, a facile and practical method to fabricate the Ni–Al–rGO intermetallic composite coating via cold spraying along with heat treatment was explored. In situ reduced graphene coated aluminum powder was conducted to effectively improve the dispersity of the graphene reinforced phase in Ni–Al intermetallic matrix coatings. The high temperature characteristic of Ni–Al–rGO composite coatings under different temperature and loading conditions were systematically investigated to reveal the wear mechanism.

## 2. Materials and Methods

### 2.1. Experiment Material

Ni, Al, and Al_2_O_3_ powders supplied by Xingrongyuan Technology Co., Ltd. (Beijing, China) were utilized for this experiment. The average particle size of the Al and Ni powders were 10 μm and 20 μm, respectively. Figure 1 illustrates the SEM morphology of the powders, where Figure 1a depicts that the aluminum powder had mainly spherical particles with a few small particles adhering to the surface of larger particles. As depicted in Figure 1b, the pure nickel powder exhibited a rough surface resembling a long pebble. The addition of Al_2_O_3_ powder aimed to enhance the powder’s flowability during the cold spray process and to improve the wear resistance of the coating. The substrate was made of 45# steel and measured 20 mm × 20 mm × 3 mm in dimensions. The steel was polished with sandpaper and subsequently cleaned using ultrasonic cleaning in a solution of alcohol and acetone to remove any oil stains before spraying.

Figure 2 depicts the schematic diagram of the high-temperature friction and wear test. For the friction and wear experiment, a pair of 6 mm diameter ZrO_2_ grinding balls with a 6 mm rotational radius were used. The experiment was conducted under dry friction conditions for a duration of 10 min, with loads of 2 N, 3 N, and 4 N, and a speed of 200 rpm. The temperatures at which the experiment was carried out were 200 °C, 400 °C, and 600 °C, respectively, in the ambient atmosphere.

### 2.2. Preparation of Coatings

Ni powder, Al powder, and Al_2_O_3_ powder were weighed according to a mass ratio of 20:6:4, and then thoroughly mixed in a three-dimensional mixer. The mixture was sprayed onto the surface of a 45# steel substrate using a supersonic low-pressure cold spraying system (423, DYMET, Obninsk, Russia) which utilized compressed air as the accelerating gas with a pressure range of 0.6–0.8 MPa and a temperature of 600 °C. The spraying speed was fixed at 300 mm/min, and the nozzle was positioned 15 mm away from the sample surface, which resulted in the deposition of a Ni–Al pre-coating. Subsequently, the pre-coating was heat-treated under an argon atmosphere in a tube furnace, with a heating rate of 10 °C/min up to 570 °C, held at this temperature for 12 h, and then cooled down to room temperature at a rate of 10 °C/min, resulting in the formation of a Ni–Al intermetallic coating.

To prepare the graphene oxide (GO) solution, an improved Hummers method was utilized [20,21,22]. In total, 20 mL of GO solution with a concentration of 2.14 mg/L was added to 200 mL of deionized water, and the mixture was sonicated and stirred for 0.5 h to obtain a homogeneous brown dispersion of graphene oxide. Then, 20 g of aluminum powder was introduced into the dispersion and stirred for 1 h. After the mixture was allowed to settle, the top layer of the clear liquid was removed, and this washing process was repeated three times using 100 mL of anhydrous ethanol each time. The resulting aluminum powder was dried in a vacuum oven at 60 °C for 10 h, yielding a powder of aluminum coated with reduced graphene oxide (Al/rGO) containing 0.2 wt.% of graphene oxide. Subsequently, Ni powder, Al/rGO powder, and Al_2_O_3_ powder were weighed according to a mass ratio of 20:6:4, thoroughly mixed in a three-dimensional mixer, and subjected to the same cold spraying and heat treatment procedures to produce a Ni–Al–rGO composite coating and a Ni–Al coating.

### 2.3. Characterization Methods for Coatings

The micro-morphology of the grinding ball’s wear surface after high-temperature wear was characterized using metallographic electron microscopy (DMi8C, Leica, Wetzlar, Germany). The cross-sectional and surface morphology of the aluminum powder coated with reduced graphene oxide and the composite coating were characterized using a field emission scanning electron microscope (SU8010, Hitachi, Tokyo, Japan). The phase composition of the reduced graphene oxide-coated aluminum powder and composite coating was analyzed using an X-ray diffractometer (D/max-γA10, Rigaku, Tokyo, Japan). The hardness values of the composite coatings were measured using a Vickers microhardness tester (MVK-H3, Akashi, Tokyo, Japan). Five points were measured at different locations on each sample, and the average value was calculated. Raman spectroscopy (LabRam HR800, Horiba, Tokyo, Japan,) was used to characterize the reduced graphene oxide (rGO)-coated aluminum powder and the wear debris from the ball milling process. The high-temperature sliding wear properties of the coatings were studied using an high-temperature friction and wear tester (MMU-5G, Hengxu, Xi’an, China). The cross-sectional area of the wear scars was measured using an laser confocal microscope (OLS4000, Olympus, Tokyo, Japan), and the wear rate was calculated.

## 3. Result and Discussion

### 3.1. Microstructure Analysis of Reduced Graphene Oxide Coated Aluminum Powder

The microstructure of aluminum powder coated with reduced graphene oxide (rGO) was characterized. Figure 3 displays the SEM morphology of 0.2 wt.% rGO-coated aluminum (Al/rGO) powder. As illustrated in Figure 3, graphene oxide was observed to be tightly wrapped around the surface of the aluminum particles in a typical wrinkled state, which is in agreement with the literature [23]. The XRD pattern of aluminum powder coated with reduced graphene oxide (rGO) is presented in Figure 4a. The characteristic diffraction peaks of aluminum are prominent, but no characteristic diffraction peaks of reduced graphene oxide were observed due to the low content of reduced graphene oxide in the sample [20,21]. To further confirm the presence of reduced graphene oxide in the composite powder, Raman spectroscopy analysis was carried out on the Al/rGO composite powder. Figure 4b illustrates the Raman spectrum of the Al/rGO composite powder. The characteristic D peak (1350 cm^−1^) and G peak (1585 cm^−1^) of reduced graphene oxide were clearly visible, indicating the presence of reduced graphene oxide in the composite powder [20,22]. In addition, compared with the as-prepared graphene oxide, the ID/IG of in situ reduced graphene exhibited a lower value, suggesting a successful fabrication of in situ reduced graphene oxide with fewer defects.

### 3.2. Observation of Composite Coating Morphology and Physical Phase Analysis

The micro-surface and cross-sectional morphologies of the Ni–Al intermetallic coating and the Ni–Al–rGO composite coating are shown in Figure 5a,b, respectively, while Figure 5c,d present the micro-surface and cross-sectional morphologies of the Ni–Al–rGO composite coating. It can be observed from Figure 5a,c that the powder particles in the composite coating are tightly bonded, and the surface exhibits an obvious fibrous and concave–convex morphology. The thickness of both coatings is about 500 μm, and the composite coating is tightly bonded to the substrate. The pore rates of the Ni–Al intermetallic coating and the Ni–Al–rGO composite coating, calculated by Image-Pro Plus image processing software, are 1.96% and 1.15%, respectively. The pore rate of the Ni–Al–rGO composite coating is lower, and the density is increased by about 41.3%. The higher density of the Ni–Al–rGO composite coating is due to the excellent thermal conductivity of graphene, which promotes the softening and plastic deformation of the coated powder particles during the spraying process. This is beneficial for the aluminum powder particles to undergo rheology and fill the pores [23].

The X-ray diffraction patterns of the Ni–Al intermetallic coating and Ni–Al–rGO composite coating are presented in Figure 6. It can be observed that the main phases of the two coatings are Ni_2_Al_3_ (JCPDS:14-0648) intermetallic, and small amounts of NiAl (JCPDS:44-1188), Ni_3_Al (JCPDS:09-0097), Ni (JCPDS:04-0850), and Al_2_O_3_ (JCPDS:82-1399). The presence of elemental Ni in small amounts is attributed to the incomplete solid-state reaction. The diffraction peak of reduced graphene oxide (rGO) was not detected in the composite coating due to its low content.

### 3.3. Composite Coating Hardness

Ni–Al intermetallic coatings and Ni–Al–rGO composite coatings were polished, and their surface microhardness was measured using an microhardness tester with a test load of 0.2 N and a time of 10 s. The hardness values were averaged from five testing points for each coating. Figure 7 shows a bar chart of the surface microhardness of the Ni–Al intermetallic coatings and Ni–Al–rGO composite coatings. As illustrated in Figure 7, the average hardness of the Ni–Al–rGO composite coating is 271 HV_0.2_, while that of the Ni–Al intermetallic coating is 261 HV_0.2_. The addition of reduced graphene oxide increases the hardness of the Ni–Al–rGO composite coating by 3.8%, as it reduces the porosity of the composite coating and enhances its hardness [24].

### 3.4. High Temperature Friction Performance Test of Composite Coating

#### 3.4.1. Friction Factors of Coatings under Different Temperatures with the Same Load

To investigate the influence of reduced graphene oxide (rGO) on the sliding friction and wear properties of Ni–Al intermetallic coatings, the friction coefficient curves of Ni–Al–rGO composite coatings and Ni–Al intermetallic coatings were obtained under a 4 N load at different temperatures, as shown in Figure 8. As depicted in Figure 8a, at a test temperature of 200 °C, the friction coefficient of 45# steel gradually increases from 0.62 to 1.24, with some fluctuations. The friction coefficient of the Ni–Al intermetallic coating remained stable at around 0.92 during the test, whereas the friction coefficient of the Ni–Al–rGO coating remained stable at around 0.85 throughout the wear process.

As shown in Figure 8b, at a test temperature of 400 °C, the friction coefficient of 45# steel smoothly increases from 0.62 to 1.22 and suddenly drops to around 1.00 at the 8th minute, then slowly rises to 1.20. The friction coefficient of the Ni–Al intermetallic coating rapidly increases from 0.40 to 0.82 within 0–2 min and slowly decreases to 0.76 within 2–10 min during the wear process. Meanwhile, the friction coefficient of the Ni–Al–rGO composite coating rapidly drops from 0.84 to 0.60 within 0–2 min and remains stable at around 0.60 within 2–10 min.

Figure 8c shows the experimental results under 600 °C conditions. The friction coefficient of the 45# steel substrate steadily increased from 0.15 to 0.86 within the first 0–5 min of the wear process. In the following 0.5 min, the friction coefficient rapidly decreased to 0.56, and then gradually increased to 0.82. The significant fluctuation in the friction coefficient of the 45# steel may be due to the dynamic response of the formation and shedding of the surface oxide film. The friction coefficient of the Ni–Al intermetallic coating remained stable at around 0.75 during the wear process, while that of the Ni–Al–rGO coating remained stable at around 0.56.

The above results indicate that, under the same load and different temperature conditions, the Ni–Al–rGO composite coating exhibits good friction-reducing properties, with a significantly lower friction coefficient compared to the 45# steel and Ni-Al intermetallic coating. The friction coefficient of the 45# steel, Ni–Al intermetallic coating, and Ni–Al–rGO composite coating all decreased with the increasing friction temperature.

#### 3.4.2. Friction Factor of the Coating under Different Loading Conditions at the Same Temperature

Figure 9 shows the friction coefficient curves under different loads at a test temperature of 400 °C. As shown in Figure 9a, when the test load is 2 N, the friction coefficient of the 45# steel substrate fluctuates greatly in the initial stage of wear. After 6 min of testing, the friction coefficient stabilizes between 1.20–1.30. The friction coefficient of the Ni–Al–rGO composite coating changes relatively steadily, starting at around 1.00 in the initial stage of testing and slowly rising to around 1.10 after approximately 4 min, where it remains relatively stable. During the wear process of the Ni–Al–rGO composite coating, the friction coefficient slowly rises from 0.38 to 0.71 in the first minute and then gradually decreases, remaining around 0.58 over the following 9 min.

Figure 9b shows the test results under a load of 3 N at 400 °C. The friction coefficient of 45# steel fluctuates greatly in the initial stage of wear, rising from 0.90 to 1.05 in the first 1.5 min, dropping to 0.91, and then continuing to rise to around 1.34 after the following 8.5 min. The friction coefficient of the Ni-Al intermetallic coating remains stable at around 0.88 during the wear process. The friction coefficient of the Ni–Al–rGO composite coating steadily decreases from 0.82 to 0.6 in the first 2 min and then remains around 0.60 in the following 8 min. The friction coefficients of the 45# steel substrate, Ni-Al intermetallic coating, and Ni–Al–rGO composite coating at 400 °C and 4 N load conditions are shown in Figure 8b.

The above results indicate that the Ni–Al–rGO composite coating exhibits good friction-reducing properties under the same temperature and different load conditions, with a significantly lower friction coefficient compared to the 45# steel and Ni-Al intermetallic coating. Under constant temperature conditions, the friction coefficients of the 45# steel and Ni–Al–rGO composite coating are less affected by load changes, while the friction coefficient of the Ni-Al intermetallic coating decreases with the increasing load.

In order to better investigate the effects of temperature and load on the friction coefficient, Table 1 presents the friction coefficients of the 45# steel, Ni-Al, and Ni–Al–rGO composite coatings in the steady-state wear stage under different working conditions. The results indicate that in high-temperature sliding friction and wear experiments conducted at 200 °C, 400 °C, and 600 °C, the friction coefficient decreases with the increasing test temperature. Moreover, the friction coefficient of the Ni–Al–rGO composite coating is significantly lower than that of the Ni-Al intermetallic coating, and the friction coefficient of the Ni-Al intermetallic coating is significantly lower than that of the 45# steel substrate. These findings demonstrate that the reduced graphene oxide (rGO) in the composite coating plays a significant role in solid lubrication [14,15], which can further enhance the high-temperature tribological properties of the Ni-Al intermetallic coating.

Under constant load conditions, the friction coefficient of the Ni–Al–rGO composite coating decreases with increasing temperature. However, under constant temperature conditions, the friction coefficient remains relatively stable with the increasing load. On the other hand, the friction coefficient of the Ni–Al intermetallic coating decreases with the increasing temperature and load, and the reduction in load has a more significant effect.

### 3.5. Coating Lubrication Mechanism Analysis

In this study, micro-morphological and Raman analyses of ZrO_2_ coated balls were conducted to investigate the tribological mechanisms of the materials used in the high-temperature wear process. Figure 10 displays the micro-morphologies of ZrO_2_ coated balls after sliding against 45# steel, Ni-Al, and Ni–Al–rGO composite coatings under 400 °C and a 3 N load. The results indicate that all the coated ball surfaces experienced varying degrees of wear with some abrasive debris accumulation on the worn surface. Of the three samples, the 45# steel had the largest wear area, while the Ni–Al–rGO composite coating had the smallest wear area, and the wear area of the Ni–Al intermetallic coating was intermediate. These results correspond well with the friction coefficients of the three samples under the same test conditions, where the friction coefficient of the Ni-Al intermetallic coating was lower than that of the 45# steel, and the friction coefficient of the Ni–Al–rGO composite coating was lower than that of the Ni-Al intermetallic coating. The further reduction in friction coefficient of the composite coating is attributed to the reduced graphene oxide (rGO) in the composite coating, which acts as an additive phase and plays a role in solid lubrication [15]. The micro-mechanism involves the solid lubrication phase forming a transfer film with low shear strength between the friction pairs, which provides lubrication and separates direct contact between the friction pairs, thereby reducing the friction coefficient of the coating. This is similar to the results obtained in previous studies [25,26] that explored the tribological characteristics of graphene/nickel–aluminum composite coatings.

To further confirm the existence of a reduced graphene oxide transfer film on the worn balls, Raman spectroscopy was performed on the balls after being rubbed against Ni–Al–rGO composite coatings under a load of 3 N and at a temperature of 400 °C. As shown in Figure 11, clear D (1326 cm^−1^) and G (1596 cm^−1^) peaks of oxidized graphene were observed on the worn surface of the balls, confirming the transfer of reduced graphene oxide from the composite coating to the worn balls during the friction and wear process [21].

Furthermore, the friction coefficients of 45# steel, Ni–Al intermetallic, and Ni–Al–rGO composite coatings decreased to varying degrees with an increase in temperature under a constant load. This phenomenon was attributed to the solid lubrication effect of the oxide film itself. As the temperature increased during the wear process, the degree of oxidation of the frictional surface increased, and the oxide film participated in the friction and wear process. Any oxide that can easily adhere to the concave and convex surface of the friction surface will reduce the frictional force on the contact surface of the friction pair [25].

### 3.6. Wear Profile and Wear Rate Rate of the Coating

Figure 12 displays the SEM morphology of the friction and wear scars of the 45# steel, Ni-Al intermetallic, and Ni–Al–rGO composite coatings under a 3 N load and at 400 °C. Figure 12a presents the SEM morphology of the wear scar of the 45# steel, which exhibits a clear and deep wear scar surface with obvious ploughing grooves and a small amount of adhesive wear. Based on this, it can be concluded that the primary wear mechanism of 45# steel under these experimental conditions is abrasive wear, accompanied by slight adhesive wear. Figure 12b shows the SEM morphology of the wear scar of the Ni-Al intermetallic coating, which is shallower than that of the 45# steel and displays obvious flaky plastic deformation with significant accumulation of wear debris at the edge of the wear scar. Therefore, it can be inferred that the wear mechanism of the coating is mainly adhesive wear, accompanied by a small amount of abrasive wear. Figure 12c displays the SEM morphology of the wear scar of the Ni–Al–rGO composite coating, which is relatively flat and shallow but exhibits some local coating delamination. Therefore, it can be concluded that the wear mechanism of the coating is only partial delamination wear. The three-dimensional wear scar morphology at the bottom of Figure 9 is in good agreement with the SEM morphology.

The coating wear rates were calculated using Equation (1) based on the cross-sectional wear track obtained by laser confocal microscopy at 400 °C and a 3 N load:(1)$$W=\frac{V}{L\cdot D}$$

The wear rate (W) is given in mm^3^ N^−1^ m^−1^, where V is the wear volume in mm^3^, L is the sliding distance in meters, and D is the loading force in Newtons. Figure 13 compares the wear rates of 45# steel, Ni–Al intermetallic, and Ni–Al–rGO composite coatings, which were determined using this method. The wear rate of the 45# steel was approximately 7.84 × 10^−9^ mm^3^ N^−1^ m^−1^, while that of the Ni–Al intermetallic coating was 5.02 × 10^−9^ mm^3^ N^−1^ m^−1^, and that of the Ni–Al–rGO composite coating was 3.52 × 10^−9^ mm^3^ N^−1^ m^−1^. The wear rate of the Ni–Al–rGO composite coating decreased by 29.9% compared with that of the Ni–Al intermetallic coating and by 55.1% compared with that of the 45# steel. These results indicate that the Ni–Al intermetallic coating has higher wear resistance than the 45# steel substrate under high-temperature conditions, and the addition of reduced graphene oxide to the Ni–Al intermetallic coating further enhances its high-temperature wear resistance [19].

## 4. Conclusions

Ni–Al–rGO composite coatings were prepared successfully using cold spray technology by in situ reducing graphene oxide to uniformly cover the surface of aluminum powder, effectively improving the dispersity of the graphene reinforcement phase in the intermetallic matrix coatings. The coated samples were subjected to heat treatment at 570 °C for 12 h to obtain Ni–Al–rGO intermetallic composite coatings.

The Ni–Al–rGO intermetallic composite coatings exhibited excellent high-temperature tribological properties. The graphene oxide formed a transfer film with low shear strength between the friction pairs, resulting in a reduction in the friction coefficient from a value of 0.9 to 0.6. The friction coefficient of the Ni–Al–rGO composite coatings decreased by 33.3% compared to the Ni–Al coatings, remaining stable during the sliding process at high temperature.

The Ni–Al–rGO intermetallic composite coatings prepared using low-pressure cold spray and heat treatment demonstrated excellent high-temperature anti-wear and anti-friction properties. Under a 3 N load and at 400 °C, the wear rate of the Ni-Al intermetallic coatings was 5.02 × 10^−9^ mm^3^ N^−1^ m^−1^, while that of the Ni–Al–rGO composite coatings was 3.52 × 10^−9^ mm^3^ N^−1^ m^−1^. The wear rate of the Ni–Al–rGO composite coatings decreased by 29.9% compared to that of the Ni-Al coatings, indicating superior high-temperature wear resistance.

## Figures and Tables

**Figure 1 materials-16-03537-f001:**
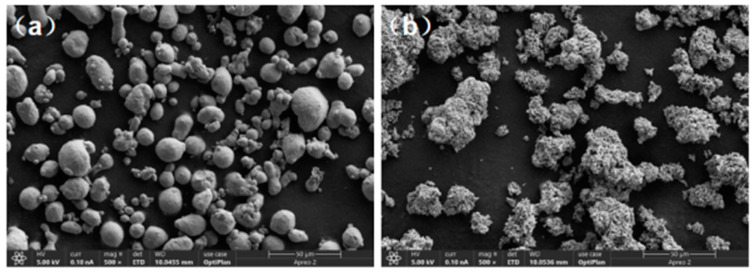
SEM of morphology powders: (**a**) Al; and (**b**) Ni.

**Figure 2 materials-16-03537-f002:**
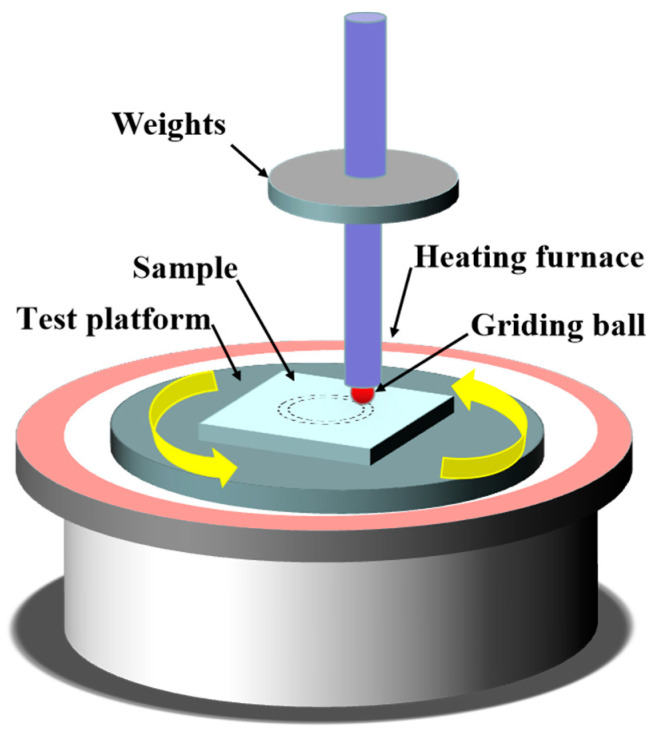
Schematic diagram of high temperature friction and wear.

**Figure 3 materials-16-03537-f003:**
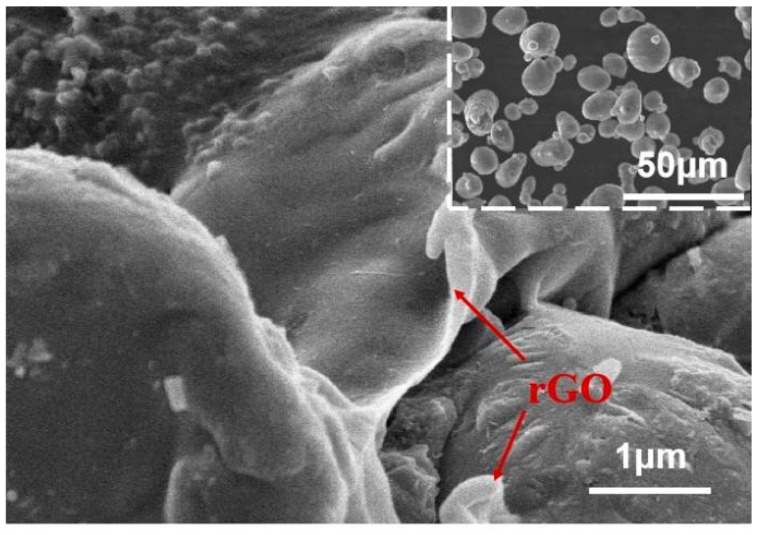
The SEM of graphene coated aluminum powder.

**Figure 4 materials-16-03537-f004:**
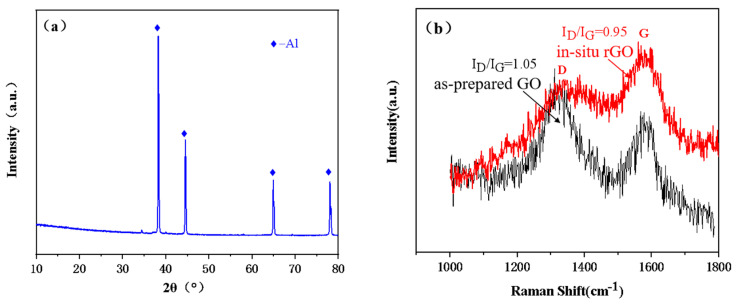
The XRD and Raman diagrams of reduced graphene oxide coated aluminum powder ((**a**) XRD Diagram of Composite Powder and (**b**) Raman Diagram of Composite Powder).

**Figure 5 materials-16-03537-f005:**
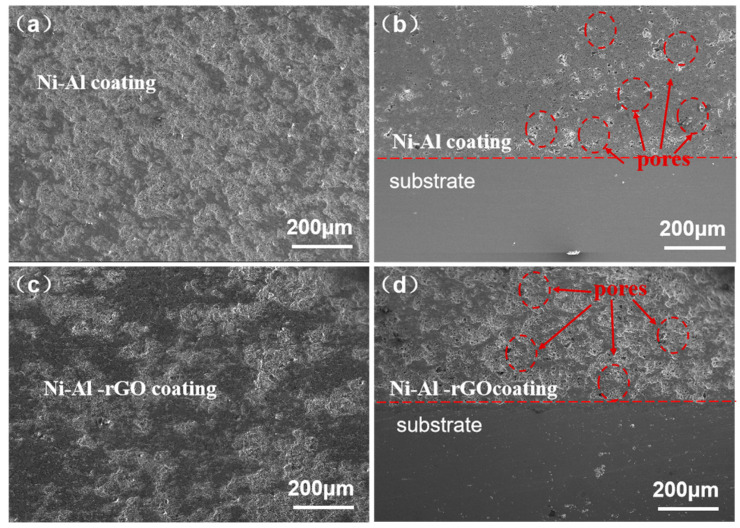
The surface and the cross-section morphology of composite coating ((**a**,**b**) are Ni–Al composite coating coatings, and (**c**,**d**) are Ni–Al–rGO composite coating).

**Figure 6 materials-16-03537-f006:**
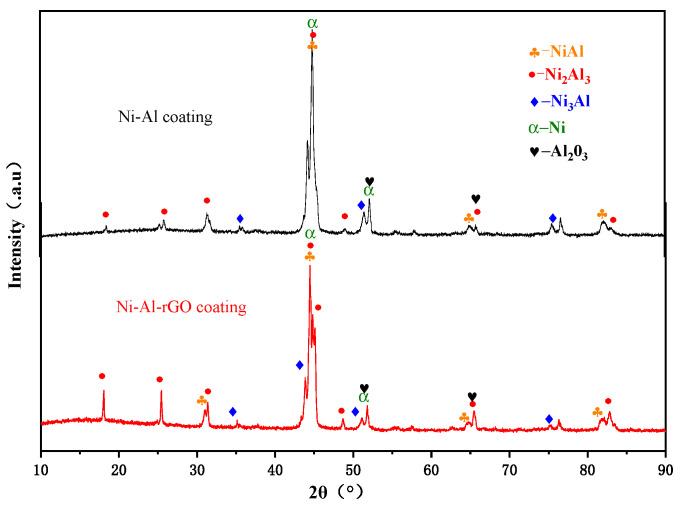
The XRD Spectrum of Ni–Al–rGO coating and Ni–Al coating.

**Figure 7 materials-16-03537-f007:**
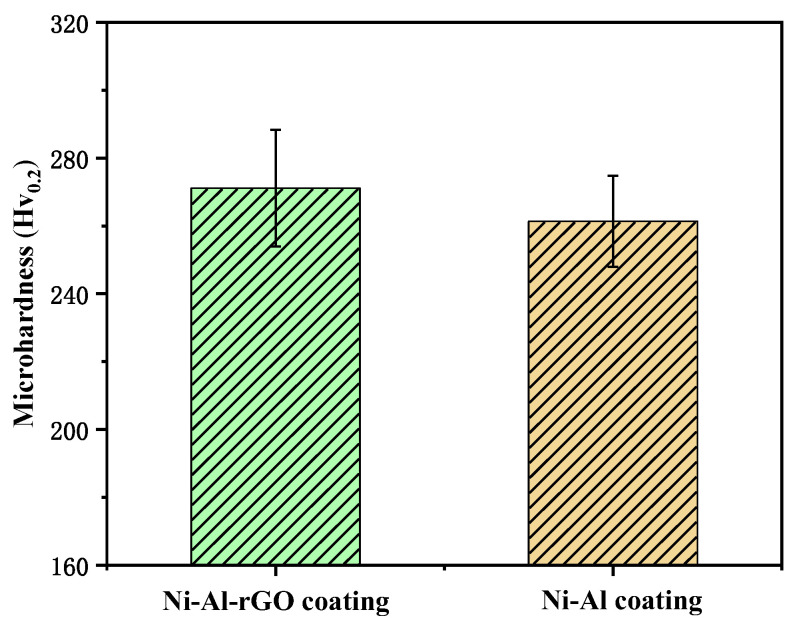
Comparison diagram of the microhardness of Ni–Al–rGO coating and Ni–Al coating.

**Figure 8 materials-16-03537-f008:**
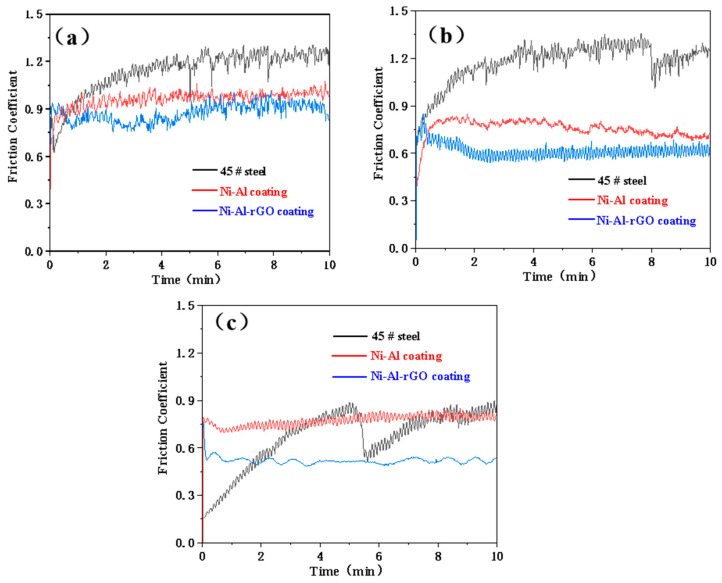
Friction coefficient of 45 # steel, NiAl and Ni–Al–rGO coating under 4 N load at different temperatures ((**a**) 200 °C, (**b**) 400 °C, (**c**) 600 °C).

**Figure 9 materials-16-03537-f009:**
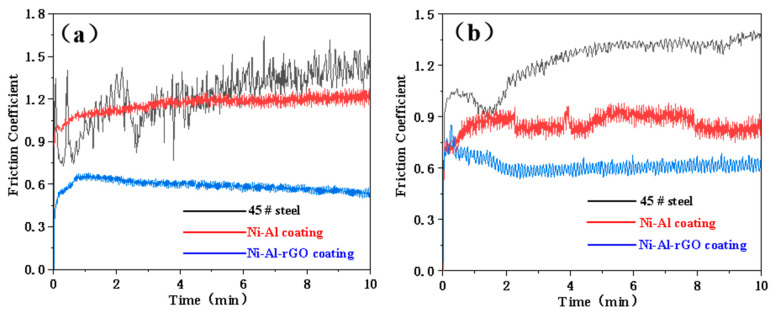
Friction coefficient of 45 # steel, Ni–Al and Ni–Al–rGO coating under different loads at 400 °C: (**a**) 2 N, (**b**) 3 N.

**Figure 10 materials-16-03537-f010:**
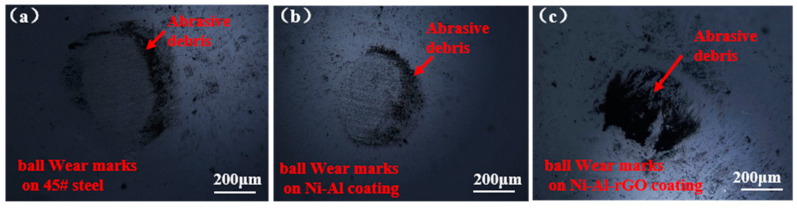
Wear microtopography of ZrO_2_ on grinding ball: (**a**) The opposite grinding pair is 45# steel; (**b**) Ni–Al coating is applied to the grinding pair; and (**c**) Ni–Al–rGO coating is applied to the grinding pair).

**Figure 11 materials-16-03537-f011:**
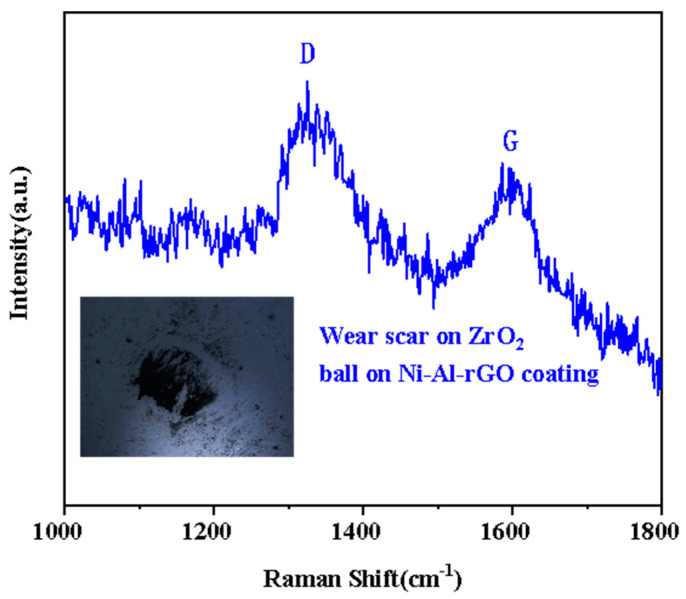
The Raman Analysis of Wear Marks of Ni–Al–rGO Composite Coating and ZrO_2_ on Grinding Ball Debris.

**Figure 12 materials-16-03537-f012:**
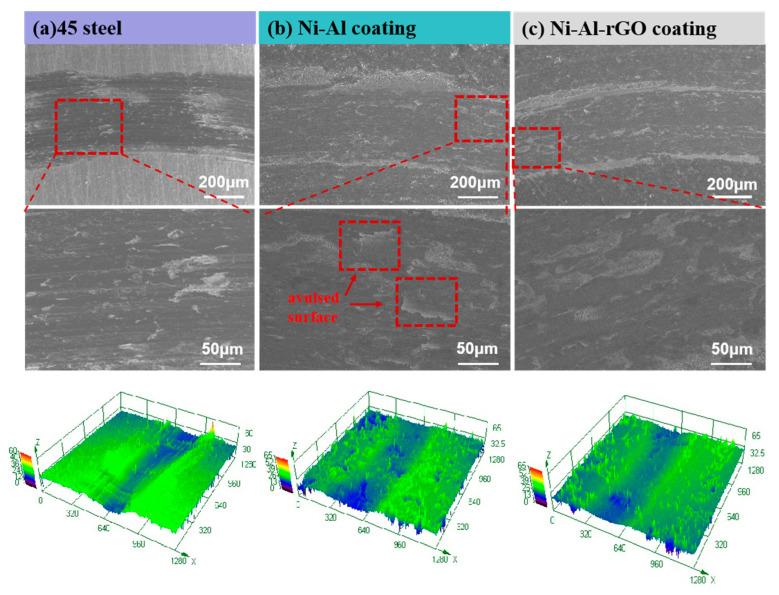
SEM and 3D morphology of wear marks on the coating surface (**a**) 45# steel; (**b**) Ni-Al coating; and (**c**) Ni–Al–rGO coating.

**Figure 13 materials-16-03537-f013:**
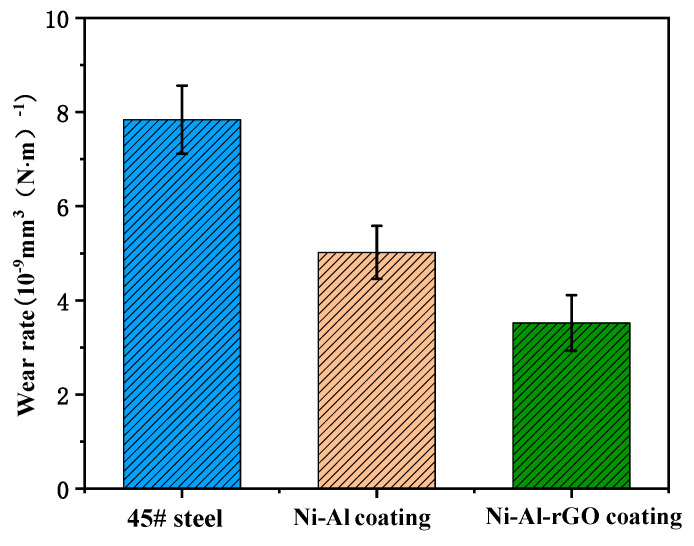
Wear rate of coating under a 3 N load at 400 °C.

**Table 1 materials-16-03537-t001:** Comparison of friction coefficient between 45 # steel, Ni–Al and Ni–Al–rGO coating under different load and temperature.

	Load	45# steel	Ni–Al	Ni–Al–rGO
Coating Temperature				
4 N, 200 °C		1.24	0.92	0.85
4 N, 400 °C		1.20	0.76	0.60
4 N, 600 °C		0.82	0.75	0.56
2 N, 400 °C		1.26	1.10	0.58
3 N, 400 °C		1.34	0.88	0.60
4 N, 400 °C		1.20	0.76	0.60

## Data Availability

Not applicable.

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
