# Peer review of "Fabrication of In Situ rGO Reinforced Ni–Al Intermetallic Composite Coatings by Low Pressure Cold Spraying with Desired High Temperature Wear Characteristics"

_materials, 2023, doi:10.3390/ma16093537_

Round 1
Reviewer 1 Report
Manuscript ID: materials-2350933
Title: Fabrication of in-situ rGO reinforced Ni-Al intermetallic composite coatings by low-pressure cold spraying with desired high-temperature wear characteristics
The respected authors described the preparation of the Ni-Al intermetallic composite coating reinforced by graphene by cold spraying along with heat treatment. The authors should address the following comments before the final decision:
• Avoid collective citations. [1-6] …
· Check the language and revise typos and grammatical errors.
• At the end of the introduction section: Explain the novelty of the study in more detail.
• It is suggested to discuss the mechanism of heat transfer in composite and FGM materials in the introduction. referring to the following papers will be useful: Doi: 10.1007/s11630-021-1517-1; 10.1007/s10973-020-09482-5.
• Enrich the conclusion section of the paper with more quantitative data from the research model.
Manuscript ID: materials-2350933
Title: Fabrication of in-situ rGO reinforced Ni-Al intermetallic composite coatings by low-pressure cold spraying with desired high-temperature wear characteristics
The respected authors described the preparation of the Ni-Al intermetallic composite coating reinforced by graphene by cold spraying along with heat treatment. The authors should address the following comments before the final decision:
• Avoid collective citations. [1-6] …
· Check the language and revise typos and grammatical errors.
• At the end of the introduction section: Explain the novelty of the study in more detail.
• It is suggested to discuss the mechanism of heat transfer in composite and FGM materials in the introduction. referring to the following papers will be useful: Doi: 10.1007/s11630-021-1517-1; 10.1007/s10973-020-09482-5.
• Enrich the conclusion section of the paper with more quantitative data from the research model.
Author Response
Ms. Ref. No.: Materials-2350933
Title: Fabrication of in-situ rGO reinforced Ni-Al intermetallic composite coatings by low pressure cold spraying with desired high temperature wear characteristics
Materials
Dear editor,
Thank you very much for your letter and the comments from the reviewers about our paper submitted to Materials (Materials-2350933). We have checked the manuscript and revised it according to reviewers’ comments. All changes in the revised manuscript have been highlighted in yellow.
Sincerely yours,
Dr. Weiqi Lian
School of Materials and Energy
Guangdong University of Technology
Guangzhou
510006
- mail:[email protected]
Reviewer1
Comments and Suggestions for Authors
Manuscript ID: materials-2350933
Title: Fabrication of in-situ rGO reinforced Ni-Al intermetallic composite coatings by low-pressure cold spraying with desired high-temperature wear characteristics
The respected authors described the preparation of the Ni-Al intermetallic composite coating reinforced by graphene by cold spraying along with heat treatment. The authors should address the following comments before the final decision:
- Avoid collective citations. [1-6] …
Answer: We are sorry for our negligence. Thanks for the practical advice. The manuscript has been carefully revised to avoid collective citations.
- Check the language and revise typos and grammatical errors.
Answer: Thanks for your advice.The whole manuscript has been carefully revised.
- At the end of the introduction section: Explain the novelty of the study in more detail.
Answer: Thanks for the precious advice from the reviewer. The end of introduction were modified to highlight the novelty of this study. The end of introduction was modified as followed:
In this study, a facile and practical method to fabricate the Ni-Al-rGO intermetallic composite coating via cold spraying along with heat treatment was explored. In-situ reduced graphene coated aluminum powder was conduct to effectively improve the dispersity of graphene reinforced phase in Ni-Al intermetallic matrix coatings.High temperature characteristic of Ni-Al-rGO composite coatings under different temperature and loading conditions were systematically investigated to reveal the wear mechanism.
- It is suggested to discuss the mechanism of heat transfer in composite and FGM materials in the introduction. referring to the following papers will be useful: Doi: 10.1007/s11630-021-1517-1; 10.1007/s10973-020-09482-5.
Answer: Thanks for your advice. The works mentioned above were added to the list of references.
- Enrich the conclusion section of the paper with more quantitative data from the research model.
Answer: Thanks for the precious advice from the reviewer. The conclusion was modified as followed:
Ni-Al-rGO composite coatings were prepared successfully using cold spray technology by in-situ reducing graphene oxide to uniformly cover the surface of aluminum powder, effectively improving the dispersity of graphene reinforcement phase in the intermetallic matrix coatings. The coated samples were subjected to heat treatment at 570°C for 12 hours to obtain Ni-Al-rGO intermetallic composite coatings.
The Ni-Al-rGO intermetallic composite coatings exhibited excellent high-temperature tribological properties.The graphene oxide formed a transfer film with low shear strength between the friction pairs, resulting in reduction of friction coefficient from a value of 0.9 to 0.6. The friction coefficient of the Ni-Al-rGO composite coatings decreased by 33.3% compared to Ni-Al coatings, remaining stable during the sliding process at high temperature.
The Ni-Al-rGO intermetallic composite coatings prepared by low-pressure cold spray and heat treatment demonstrated excellent high-temperature anti-wear and anti-friction properties. Under 3N load and at 400°C, the wear rate of Ni-Al intermetallic coatings was 5.02×10-9 mm3N-1m-1, while that of Ni-Al-rGO composite coatings was 3.52×10-9 mm3N-1m-1. The wear rate of the Ni-Al-rGO composite coatings decreased by 29.9% compared to that of Ni-Al coatings, indicating superior high-temperature wear resistance.
Reviewer 2 Report
Very thorough characteristics of NiAl-rGO coatings along with their wear mechanisms.
I would like to mention that composite coatings based on Ni-Al metallic compounds are very interesting construction materials with a wide range of engineering applications in many technologically important areas. The spectrum of their properties determines their suitability for work at elevated temperatures, which was effectively presented in the article. Also the technique used for their production has many advantages - in terms of ecology and quality. It is worth emphasizing that the methodology used, and in particular the characteristics of the microstructure, is appropriate. In the future, it can be enriched with indentation tests (hardness, Young's modulus and fracture toughness) and adhesion tests (in the scratch test).Author Response
Ms. Ref. No.: Materials-2350933
Title: Fabrication of in-situ rGO reinforced Ni-Al intermetallic composite coatings by low pressure cold spraying with desired high temperature wear characteristics
Materials
Dear editor,
Thank you very much for your letter and the comments from the reviewers about our paper submitted to Materials (Materials-2350933). We have checked the manuscript and revised it according to reviewers’ comments. All changes in the revised manuscript have been highlighted in yellow.
Sincerely yours,
Dr. Weiqi Lian
School of Materials and Energy
Guangdong University of Technology
Guangzhou
510006
Reviewer2
Comments and Suggestions for Authors
Manuscript ID: materials-2350933
Very thorough characteristics of NiAl-rGO coatings along with their wear mechanisms.
I would like to mention that composite coatings based on Ni-Al metallic compounds are very interesting construction materials with a wide range of engineering applications in many technologically important areas. The spectrum of their properties determines their suitability for work at elevated temperatures, which was effectively presented in the article. Also the technique used for their production has many advantages - in terms of ecology and quality. It is worth emphasizing that the methodology used, and in particular the characteristics of the microstructure, is appropriate. In the future, it can be enriched with indentation tests (hardness, Young's modulus and fracture toughness) and adhesion tests (in the scratch test).
Answer:
Thanks for the reviewer’s precious advice and approvals.
Reviewer 3 Report
This paper deals with the development of in-situ rGO reinforced Ni-Al intermetallic composite coatings. There are a lot of characterizations, and the development is interesting. But the manuscript looks still a bit drafty, the paper needs major revision before publication. Here are some comments:
- The introduction misses a bit of context and application of the coating.
- The title of the section 2 is strange, usually, it is Experimental section or Materials and Methods.
- The section 2.3 is very short, more details is needed.
- There are a lot of typos along the text and the English needs to be checked.
- P3 Hummers method needs reference.
- For XRD, the JPCD references are missing.
- For Raman peaks, references are missing too.
- In Table 1, the title of the first column can be only “Load, Temperature” because we understand that it is the coatings values that are detailed.
- Comparison with similar coatings is missing.
- The conclusion is very short and did not give the highlights of this work, it needs to be rewritten.
There are a lot of typos along the text and the English needs to be checked.
Author Response
Ms. Ref. No.: Materials-2350933
Title: Fabrication of in-situ rGO reinforced Ni-Al intermetallic composite coatings by low pressure cold spraying with desired high temperature wear characteristics
Materials
Dear editor,
Thank you very much for your letter and the comments from the reviewers about our paper submitted to Materials (Materials-2350933). We have checked the manuscript and revised it according to reviewers’ comments. All changes in the revised manuscript have been highlighted in yellow.
Sincerely yours,
Dr. Weiqi Lian
School of Materials and Energy
Guangdong University of Technology
Guangzhou
510006
- mail:[email protected]
Reviewer3
Comments and Suggestions for Authors
Manuscript ID: materials-2350933
Comments and Suggestions for Authors
This paper deals with the development of in-situ rGO reinforced Ni-Al intermetallic composite coatings. There are a lot of characterizations, and the development is interesting. But the manuscript looks still a bit drafty, the paper needs major revision before publication. Here are some comments:
- The introduction misses a bit of context and application of the coating.
Answer: Thanks for your advice.The introduction has been modified as followed:
Recent studies have demonstrated that aluminum-rich alloy coatings exhibit excellent high-temperature oxidation resistance and anti-wear properties [1-2], making them widely applied in the aerospace and automobile industries[3]. Traditional high-temperature oxidation-resistant coatings mainly rely on ceramic coatings such as SiO2 and Al2O3. However, their high hardness makes them prone to peeling and cracking in the cold state, despite performing well at high temperatures.
- The title of the section 2 is strange, usually, it is Experimental section or Materials and Methods.
Answer: We are sorry for our negligence. Thanks for the practical advice. The title of the section 2 was revised to Materials and Methods.
- The section 2.3 is very short, more details is needed.
Answer: Thanks for the practical and precious advice. The section 2.3 was modified as followed:
The micro-morphology of the grinding ball's wear surface after high-temperature wear was characterized using metallographic electron microscopy (OM, Leica, DMi8C). The cross-sectional and surface morphology of the aluminum powder coated with reduced graphene oxide and the composite coating were characterized using a field emission scanning electron microscope (SEM, Hitachi SU8010). The phase composition of the reduced graphene oxide-coated aluminum powder and composite coating was analyzed using an X-ray diffractometer (XRD, Rigaku, Ultima-â…£).The hardness values of the composite coatings were measured using an Akashi MVK-H3 Vickers microhardness tester. Five points were measured at different locations on each sample, and the average value was calculated. Raman spectroscopy (Raman, LabRam, HR800) was used to characterize the reduced graphene oxide (rGO)-coated aluminum powder and the wear debris from the ball milling process. The high-temperature sliding wear properties of the coatings were studied using an MMU-5G high-temperature friction and wear tester. The cross-sectional area of the wear scars was measured using an Olympus-OLS4000 laser confocal microscope, and the wear rate was calculated.
- There are a lot of typos along the text and the English needs to be checked.
Answer: Thanks for your advice.The whole manuscript has been carefully revised.
- P3 Hummers method needs reference.
Answer: Thanks for your advice. The reference of Hummers method was added as followed:
To prepare the graphene oxide (GO) solution, an improved Hummers method was utilized [20-22].
- For XRD, the JPCD references are missing.
Answer: Thanks for your advice. The references of XRD were added as followed:
The X-ray diffraction patterns of Ni-Al intermetallic coating and Ni-Al-rGO composite coating are presented in Figure 6. It can be observed that the main phases of the two coatings are Ni2Al3 (JCPDS:14-0648) intermetallic, and small amounts of NiAl (JCPDS:44-1188), Ni3Al (JCPDS:09-0097), Ni (JCPDS:04-0850), and Al2O3 (JCPDS:82-1399). The presence of elemental Ni in small amounts is attributed to the incomplete solid-state reaction. The diffraction peak of reduced graphene oxide (rGO) was not detected in the composite coating due to its low content.
- For Raman peaks, references are missing too.
Answer: Thanks for your advice. The references of Raman peaks were added as followed:
To further confirm the presence of reduced graphene oxide in the composite powder, Raman spectroscopy analysis was carried out on the Al/rGO composite powder. Figure 4(b) illustrates the Raman spectrum of the Al/rGO composite powder. The characteristic D peak (1350 cm-1) and G peak (1585 cm-1) of reduced graphene oxide were clearly visible, indicating the presence of reduced graphene oxide in the composite powder [20, 22]. In addition, compared with the as-prepared graphene oxide, the ID/IG of in-situ reduced graphene exhibited a lower value, suggesting a successful fabrication of in-situ reduced graphene oxide with fewer defects.
- In Table 1, the title of the first column can be only “Load, Temperature” because we understand that it is the coatings values that are detailed.
Answer: Thanks for your advice. The title of the first column in table 1 was revised to “Load, Temperature”.
- Comparison with similar coatings is missing.
Answer: Thanks for the precious and practical advice from the reviewer. Generally, it is hard to compare the efficiency with literature survey. And the highlights are concluded as followed:
1.The motivation and innovation of the research are to provide a facile way to fabricate Ni-Al-rGO intermetallic composite coatings via cold spraying along with heat treatment. The conventional ways to obtain ntermetallic composite coatings need at least two or more steps.
- The addition of graphene reinforcement phase is an in-situ way via
graphene oxide coated aluminum powder along with subsequent thermal reduction. This method could effectively avoid the agglomeration of graphene in intermetallic matrix coatings.
- The high temperature wear mechanism of Ni-Al intermetallic coating was previous investigated by many researches. In this study, the authors mainly focus on the wear mechanism and influence of rGO in the composite coatings.
Based on the reasons mentioned above, this research is for providing a new way for fabricating in-situ Ni-Al-rGO intermetallic composite coatings. Comparing its efficiency with other methods is not the motivation and purpose of this research.
- The conclusion is very short and did not give the highlights of this work, it needs to be rewritten.
Answer: Thanks for your advice. The conclusion was revised as followed:
Ni-Al-rGO composite coatings were prepared successfully using cold spray technology by in-situ reducing graphene oxide to uniformly cover the surface of aluminum powder, effectively improving the dispersity of graphene reinforcement phase in the intermetallic matrix coatings. The coated samples were subjected to heat treatment at 570°C for 12 hours to obtain Ni-Al-rGO intermetallic composite coatings.
The Ni-Al-rGO intermetallic composite coatings exhibited excellent high-temperature tribological properties.The graphene oxide formed a transfer film with low shear strength between the friction pairs, resulting in reduction of friction coefficient from a value of 0.9 to 0.6. The friction coefficient of the Ni-Al-rGO composite coatings decreased by 33.3% compared to Ni-Al coatings, remaining stable during the sliding process at high temperature.
The Ni-Al-rGO intermetallic composite coatings prepared by low-pressure cold spray and heat treatment demonstrated excellent high-temperature anti-wear and anti-friction properties. Under 3N load and at 400°C, the wear rate of Ni-Al intermetallic coatings was 5.02×10-9 mm3N-1m-1, while that of Ni-Al-rGO composite coatings was 3.52×10-9 mm3N-1m-1. The wear rate of the Ni-Al-rGO composite coatings decreased by 29.9% compared to that of Ni-Al coatings, indicating superior high-temperature wear resistance.
Round 2
Reviewer 3 Report
The comments were addressed, the paper can be published.
Minor editing of English language required